# Lightweight wavelet-CNN tea leaf disease detection

**Jing Yang**, **GaoJian Xu**\*, **MengDao Yang, ZhengPei Lin**

School of Information and Artificial Intelligence, Anhui Agricultural University, Hefei, Anhui, China

\* xugj@ahau.edu.cn

## Abstract

Tea diseases can significantly impact crop yield and quality, necessitating accurate and efficient recognition methods. This study presents WaveLiteNet, a lightweight model designed for tea disease recognition, addressing the challenge of inadequate disease feature extraction in existing approaches. By integrating 2D discrete wavelet transform (DWT) with MobileNetV3, the model enhances noise suppression and feature extraction through an adaptive thresholding strategy in the 2D DWT. The extracted frequency-domain features are fused with depth features from the Bneck structure, enabling a more comprehensive representation of disease characteristics. To further optimize feature extraction, a convolutional block attention module (CBAM) is incorporated within the Bneck structure, refining the network's ability to assign optimal weights to feature channels. A focal loss function also replaces traditional cross-entropy loss to mitigate sample category imbalance, improving recognition accuracy across varying distributions. Experimental results show that WaveLiteNet achieves a 98.70% recognition accuracy on five types of tea leaf diseases, with a parameter count of $3.16 \times 10^6$. Compared to MobileNetV3, this represents a 2.15 percentage point improvement in accuracy while reducing the parameter count by 25.12%. These findings underscore WaveLiteNet's potential as a highly efficient and lightweight real-time crop disease recognition solution, particularly in resource-constrained agricultural environments.

## Introduction

Diseases represent one of the most critical factors negatively impacting the yield and quality of tea. In 2021, China's tea plantations reached 3,308,000 hm², marking a 2.83% increase compared to 2020. Despite this expansion, tea production continues to face significant challenges from diseases such as leaf blight, which account for an estimated annual yield reduction of approximately 20% [1], leading to severe social and economic losses [2]. As such, the accurate and rapid identification of tea diseases is vital for mitigating these detrimental effects on yield and quality.

**Data availability statement:** All relevant data are within https://github.com/jingyang-create/tea-sickness.

**Funding:** This study was funded by a contract between my supervisor and Anhui Xiansen Green Food Co., Ltd., titled "Application Demonstration of AI Combined with Agricultural IoT Visual Algorithms for Poultry Monitoring System," contract number kj24530. The funders had no role in study design, data collection and analysis, decision to publish, or preparation of the manuscript.

In recent years, the adoption of machine learning and deep learning techniques for tea disease recognition has grown substantially, with increasing emphasis on severity estimation and deep feature extraction [3–7]. Traditional machine-learning approaches, however, often rely on extensive image segmentation and manual feature extraction. Due to the complex and variable nature of crop spot patterns, these methods are prone to segmentation inaccuracies, resulting in low robustness and poor generalization performance [8,9]. To overcome these limitations, researchers have increasingly turned to deep learning models, which have demonstrated significant improvements in recognition accuracy [10–12].

Several lightweight models have been proposed to address the practical challenges of deploying deep learning models in resource-constrained environments. For instance, Hu et al. [13] developed the MergeModel by integrating two convolutional neural networks (CNNs), achieving $1.6 \times 10^9$ floating-point operations (FLOPs) for small-sample tea disease recognition. Zimao Li et al. [14] introduced the SE-DenseNet-FL model based on DenseNet, which achieved recognition accuracy of 92.66% across five disease types, including tea white star disease, tea verticillium disease, and tea coal disease. Additionally, lightweight model that has also been proposed for other crops, such as tomato disease detection using hybrid deep learning frameworks [15], providing useful references for the development of efficient tea disease identification models. While these models significantly improve recognition performance, they remain constrained by challenges such as computational cost, high parameter counts, and difficulties in addressing complex backgrounds or imbalanced datasets.

MobileNet, as a widely used lightweight neural network, has been employed as a base model in recognition tasks to reduce parameter count and computational demands [16–18]. By integrating attention mechanisms and improving model structures, researchers have further optimized MobileNet for enhanced recognition performance with fewer resources [19–23]. However, existing lightweight models, including SE-DenseNet-FL and TealeafNet, often struggle with high-precision classification due to limited datasets, sparse disease spot distributions, and interference caused by complex image backgrounds.

To address these issues, this paper introduces WaveLiteNet, a novel lightweight model that combines frequency-domain and depth feature extraction to achieve superior recognition performance. Unlike previous models, WaveLiteNet integrates the 2D discrete wavelet transform (DWT) with MobileNetV3, leveraging the multi-resolution analysis capabilities of wavelets for improved feature extraction in complex scenarios. An adaptive thresholding design in 2D DWT dynamically adjusts the threshold based on data characteristics, effectively distinguishing between signal and noise while avoiding excessive or insufficient denoising. Furthermore, the original attention module, SENet [24] in MobileNetV3, is replaced with the convolutional block attention module (CBAM) [25], enabling more targeted learning of feature information and enhancing recognition accuracy [26–28].

The proposed WaveLiteNet model was validated on datasets comprising five categories of tea diseases: *Anthracnose*, *Pseudocercospora*, *Cercospora Leaf*

*Spot*, *Blister Blight*, and *Leaf Blight or Brown Blight of Tea*. Experimental results demonstrate that WaveLiteNet surpasses SE-DenseNet-FL, TealeafNet, and other existing lightweight models in recognition accuracy while significantly reducing the parameter count, providing an effective solution for real-world applications of tea disease recognition in resource-constrained environments.

## Materials and methods

### Experimental data

**Data sample collection.** Tea disease images were collected from the Tea Experiment and Demonstration Station at Anhui Agricultural University in Hefei City, Anhui Province, and tea gardens in Shitai County, Chizhou City, Anhui Province. The collection took place from October 2 to October 9, 2023. Images of *Anthracnose*, *Pseudocercospora*, *Cercospora Leaf Spot*, *Blister Blight*, *Leaf Blight or Brown Blight of Tea* on Wan Nong 95 and Long Jing 43 cultivars were captured using an iPhone 14 Pro Max. The characteristics of each disease type are presented in Table 1. Images were taken from multiple angles and in various environments to enhance the diversity of training samples. Examples of the collected image samples are displayed in Fig 1, with a total of 3,260 samples gathered under the guidance of the chief expert team from the Anhui Tea Industry Technology System and plant protection specialists.

**Preprocessing and dataset.** The data samples were cropped to minimize the impact of complex backgrounds on feature extraction. This involved extracting the outer rectangular box of the target in each image, followed by a cut-and-expand operation to fill the image with 0-pixel values, standardizing the size to $224 \times 224 \times 3$. The same leaf blade can exhibit significant variations in size, position, orientation, and lighting under different shooting conditions. Data augmentation techniques were applied to enhance the network model's generalization ability. Certain areas of *Pseudocercospora* spots may appear ruddy due to variations in light and angle, leading to blurriness when mixed with the original brown color.

Additionally, the irregular sizes of tea leaf spots, the varying clarity of images under different lighting, and the presence of rain and mud can introduce noise. Thus, five data enhancement methods were employed: 1) blurring, 2) scaling, 3) rotation, 4) lightening and darkening, and 5) adding noise.

The dataset was partitioned into training, validation, and test sets with a ratio of 6:2:2. Data augmentation was applied exclusively to the training set to improve the model's generalization ability, while the validation and test sets were preserved in their original form. This ensured that the validation and test sets remained independent benchmarks for evaluating model performance, free from any influence of augmented data. Table 1 provides an overview of the number of samples before and after data augmentation.

**Table 1. Diseases' characteristics in tea leaf.**

| Diseases | Characteristics | Amounts | |
|---|---|---|---|
| | | Original image | Enhanced image |
| *Anthracnose* | The spots appear semi-circular or irregular, initially manifesting as watery yellow-brown dots that gradually expand to a burnt-yellow color. | 638 | 3828 |
| *Pseudocercospora* | They start as yellowish-brown and watery, eventually developing into nearly round or irregular brown spots. | 582 | 3492 |
| *Cercospora Leaf Spot* | The initial brown dots may be partially stained with muddy images resembling them, later enlarging into round spots with a sunken center. | 709 | 4254 |
| *Blister Blight* | Initially, there are light yellow watery spots, with the middle of the abaxial surface raised like a bun. In the later stages, these spots shrivel and turn into brownish, withered areas. | 679 | 4074 |
| *Leaf Blight or Brown Blight of Tea* | The spots begin as light brown and then expand into irregular patches across half or all of the leaves, eventually becoming reddish-brown. | 652 | 3912 |

## Methods

**Fusion of frequency domain and deep features.** Tea disease images collected in natural environments often feature unstructured, overlapping, and complex backgrounds, including light shadows, weeds, and branches, which can lead to misrecognition. The 2D discrete wavelet transform (DWT) reduces the influence of noise, mitigates other interfering factors, and emphasizes detailed information in tea disease images through its multi-resolution decomposition capabilities. Furthermore, by processing the RGB channels of the image separately, the color information of the tea leaf disease images is preserved. Thus, employing 2D DWT facilitates extracting higher-order abstract features essential for distinguishing between tea leaf diseases, enhancing the model's recognition performance.

The threshold value selection is critical when utilizing 2D DWT for image denoising. A fixed threshold may result in over-denoising or under-denoising, losing essential details, or retaining excessive noise. This paper integrates the Visu-Shrink method with 2D DWT for image denoising, effectively preserving image details.

The 2D DWT is a mathematical technique that decomposes images into frequency subbands. These standards include low-frequency components, which capture the image's overall structure and smoother regions, and high-frequency components, which contain details such as edges and textures. This decomposition enables targeted processing of specific frequency ranges, making it particularly effective for denoising and feature extraction tasks.

The VisuShrink thresholding method effectively removes noise from the high-frequency subbands while preserving essential details. This approach calculates an adaptive threshold value based on the statistical properties of the wavelet

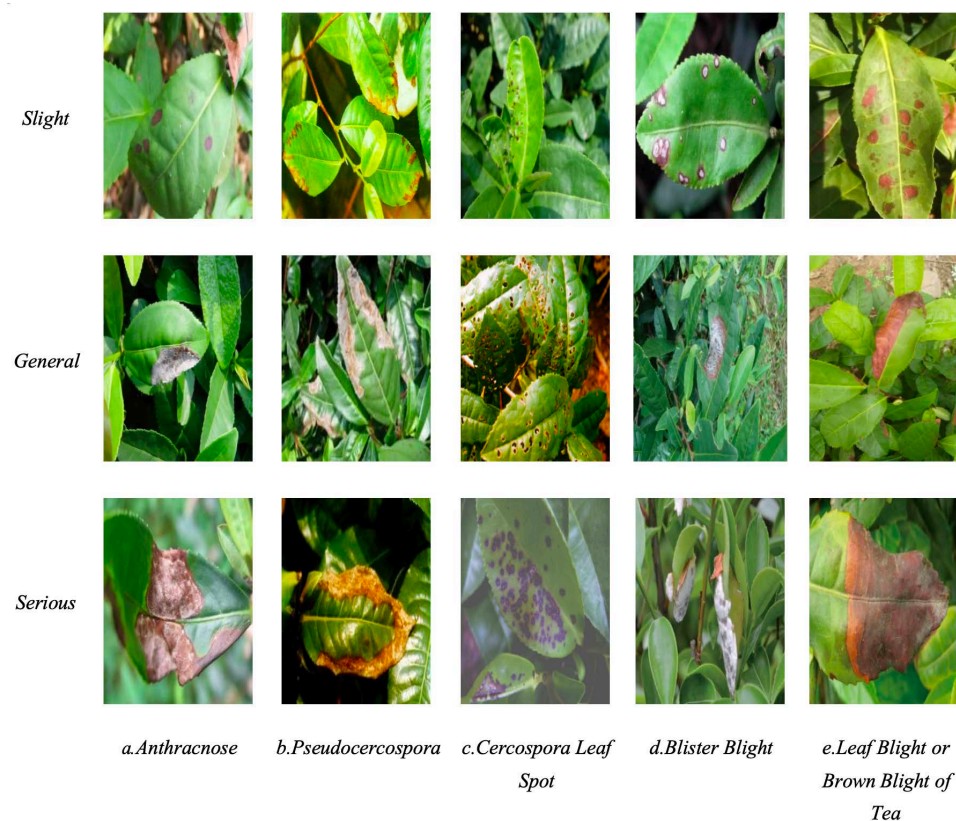

**Fig 1. Disease samples of tea leaf.**

coefficients. Using the robust median measure, the process begins by estimating the noise level in the high-frequency subbands (*cH*, *cV*, and *cD*). The noise standard deviation ($\sigma$) is determined using the following equation:

$$\sigma = \frac{median\,(|d|)}{0.6745}$$

(1)

Where *d* denotes the wavelet coefficients from the highest frequency subband. the constant 0.6745 ensures that the median value is appropriately scaled under the assumption of Gaussian noise.

Based on this noise estimated, The VisuShrink threshold (*T*) is t computed using the following formula:

$$T = \sigma\sqrt{2log(n)}$$

(2)

Where *n* represents the size of the wavelet coefficients, The threshold *T* serves as a critical boundary to differentiate between coefficients representing noise and those containing meaningful signal information. Larger thresholds effectively suppress weaker, noise-like components while retaining stronger coefficients associated with significant features.

A soft thresholding function is applied to refine the wavelet coefficients. This function shrinks coefficients with small magnitudes toward zero while leaving larger coefficients relatively unaffected. It is defined as follows:

$$d_{new} = sgn(d) \cdot max\,(|d| - T, 0)$$

(3)

Where $d_{new}$ represents the updated coefficients, *sgn*(*d*) preserves the original sign of the coefficients, and the max function ensures that coefficients smaller than the threshold *T* are set to zero.

After thresholding, the denoised image is reconstructed using the inverse 2D DWT. This step combines the processed high-frequency coefficients with the original low-frequency components, resulting in a cleaner image with reduced noise but preserved essential details. The process of specific DWT is shown in Fig 2.

In this paper, the Haar wavelet is employed to perform four layers of wavelet decomposition, as illustrated in Fig 3. Post-decomposition, most noise resides in the high-frequency com, and subsequent wavelet decompositions target the low-frequency components, progressively suppressing noise. The designed method for frequency domain and depth feature fusion is depicted in Fig 4, with the input being the feature matrix formed from the concatenation of the upper-level frequency domain and depth features.

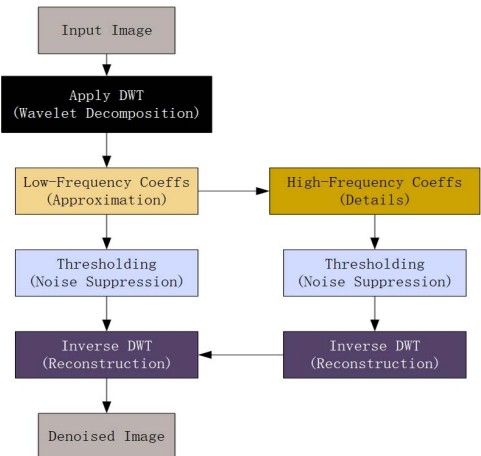

**Fig 2. The process of DWT.**

During the extraction of depth features, spatial domain features are derived from the input feature map through multiple Bneck structures. The feature matrix is halved during frequency domain feature extraction, necessitating adjustments to the spatial domain features using a Bneck structure with a stride of 2 to ensure uniform dimensions. Frequency domain features are extracted from one low-frequency feature, and three high-frequency features are obtained from the previous layer of wavelet decomposition via 2D DWT. Since the arbitrary frequency feature matrix captures only part of the disease information, it is channel-stacked to form a multi-channel feature matrix. The number of channels is aligned with those used in convolution to ensure equal weighting of frequency domain and depth features within the network. Finally, the feature maps of identical sizes from both branches are combined through channel superimposition to complete the frequency and depth features fusion.

**Tea disease overall recognition model.** Practical applications necessitate a lightweight tea disease recognition model that maintains high accuracy. This paper employs MobileNetV3 as the base model, utilizing frequency domain and depth feature fusion for effective feature extraction while incorporating the Convolutional Block Attention Module (CBAM) to optimize the SENet in the base model. The resulting network model is illustrated in Fig 5.

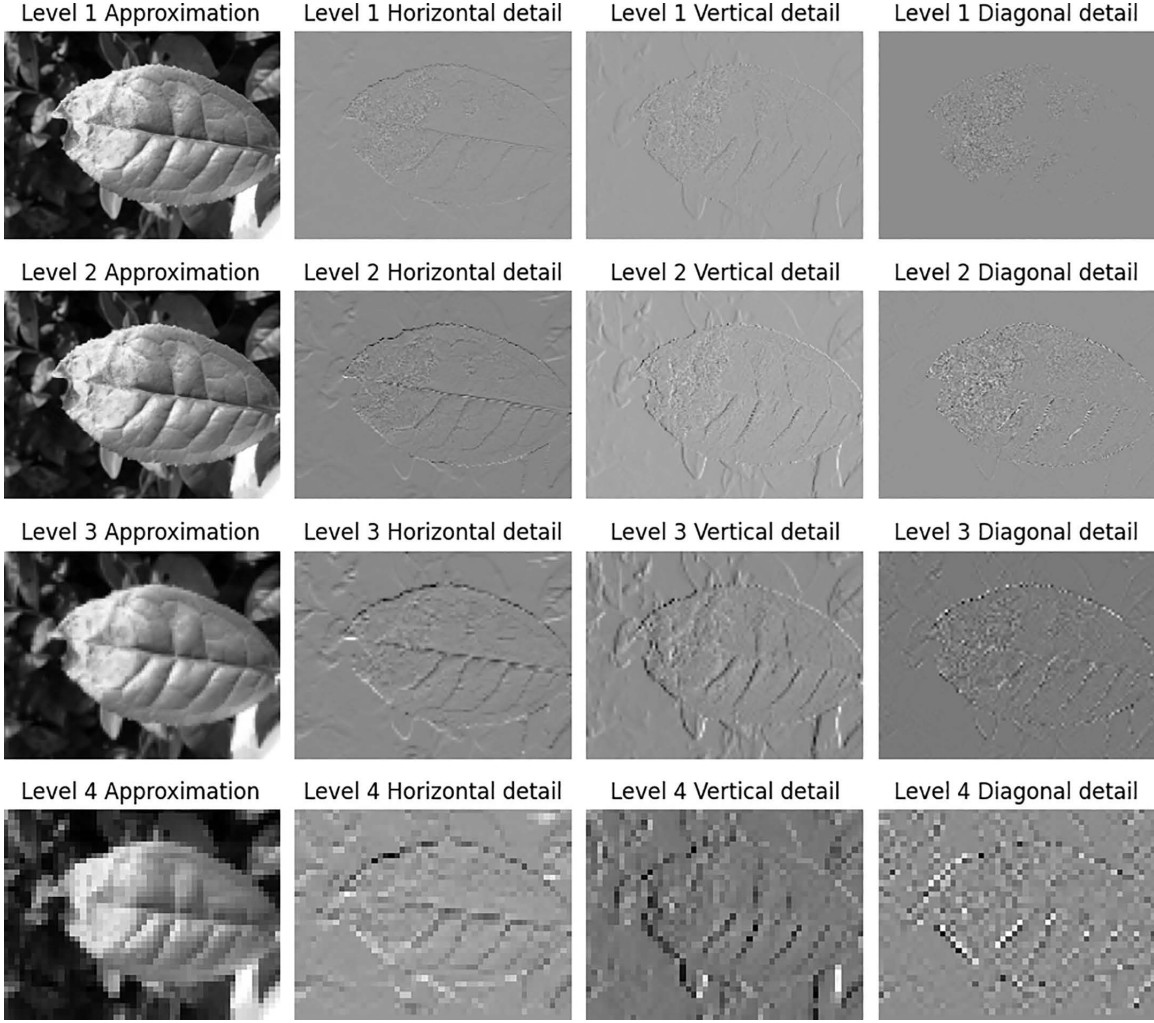

**Fig 3. Results of 4 level 2D DWT.**

The model inputs a 224×224×3 RGB image, producing outputs corresponding to five tea disease categories. The MobileNetV3 model includes multiple down-sampling processes, gradually reducing the feature map size as the network deepens. Each application of 2D DWT also contributes to reducing the width and height of the frequency domain feature matrix. Given the quantitative relationship between the frequency domain and depth feature channels, the fusion of 2D DWT with spatial domain features enriches the extracted disease features, particularly when the size of the feature matrix is reduced.

The inverse residual structure, as adopted in MobileNet-v2 [29], effectively reduces memory consumption during model inference while retaining rich feature representations, making it well-suited for deployment in resource-constrained environments. Building upon this design, the proposed model employs the h-swish activation function instead of the swish nonlinear function used in the base model. Although the swish function enhances accuracy, it incurs significant computational costs in embedded environments, mainly due to the expensive computation of the sigmoid function on mobile devices. The h-swish activation function addresses the computational and derivation complexity associated with the sigmoid function while preserving the diversity of feature information and enhancing the expressiveness of disease features. The formula for the h-swish activation function is as follows:

$$h - swish(x) = x \cdot \frac{Relu6(x+3)}{6} \tag{4}$$

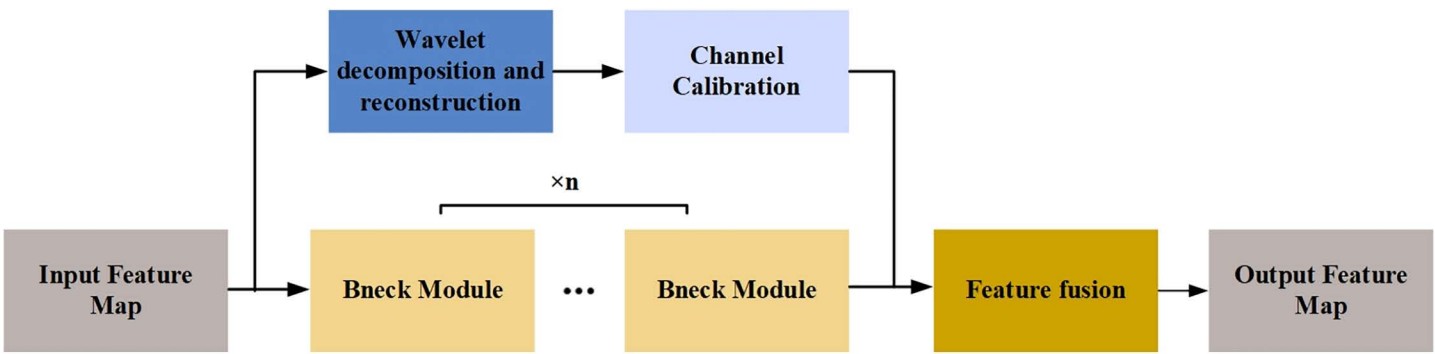

**Fig 4. Fusion of frequency and deep features.**

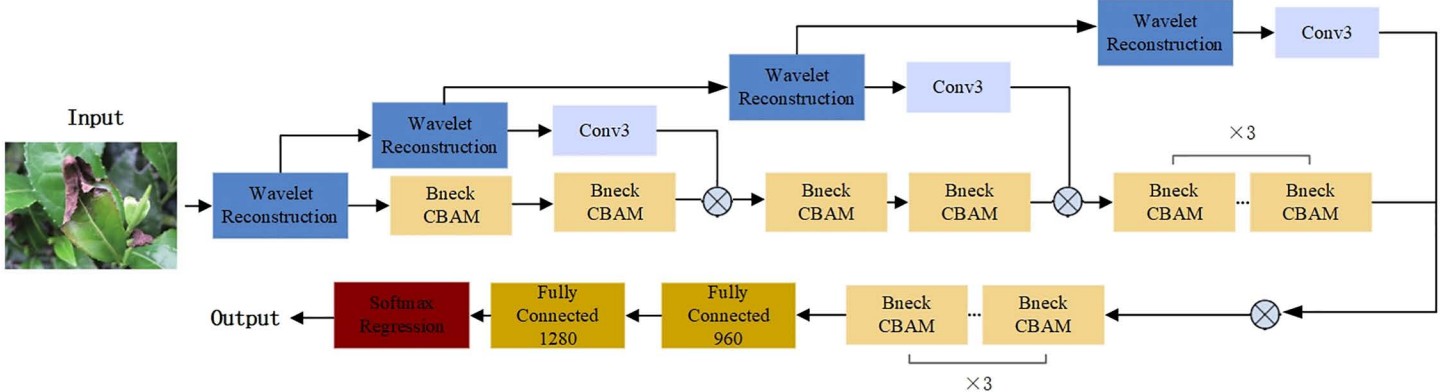

**Fig 5. The network structure of WaveLiteNet.**

Where *Relu6* is defined as $Relu6 = min(max(0, x), 6)$.

The optimized Bneck structure is illustrated in Fig 6, where the input features undergo processing through the inverse residual structure. Dimensionality is increased using a 1×1 convolution, while dimensionality reduction is achieved via a 3×3 separable convolution. Feature weights are adjusted by the Convolutional Block Attention Module (CBAM), and the h-swish activation function is employed to enhance the network's accuracy.

To determine the optimal base model structure while minimizing the number of parameters, the Bneck structure of MobileNetV3 is configured as (2, 2, 3, 6, 22,3,6) based on the input feature map size. The parameter size is primarily influenced by the number of Bneck structures in the last two segments, with four models set to (2, 2, 3, 2, 1), (2, 2, 3, 2, 2), (2, 2, 3, 4, 1)(1–4), and (2, 2, 3, 4, 22–4), respectively. The performance of these models is evaluated using accuracy, precision, recall, and F1-score as comparison metrics [30]. Practical applications require a balance between the number of parameters and model recognition speed; hence, the parameters and FLOPs of the models were compared, leading to the selection of the (2, 2, 3, 2, 11–3) structure for fusion with frequency domain features after validation.

Color is crucial for distinguishing different diseases. To preserve color features, the RGB channels of the input image undergo one-level wavelet decomposition, reconstruction, and adjustment via 3×3 convolution, resulting in a 112×112×16 feature map. This feature map is then upscaled and downscaled using Bneck with a stride of 1 to enhance model convergence while minimizing parameters, ultimately adjusting the feature map to 56×56×24 using Bneck with a stride of 2. Simultaneously, a 12-channel feature map is generated through two-stage wavelet decomposition and reconstruction, with Channel calibrations made through convolution to complete the fusion of frequency domain and depth features at layer 1.

Given that *Leaf Blight or Brown Blight of Tea* and *Pseudocercospora* exhibit similar lesion states and features, CBAM assigns different weights to feature layers during spatial feature extraction, effectively reducing feature redundancy and enhancing the model's ability to learn subtle differences in lesions. The resulting 56×56×48 feature map serves as input for the next layer. In the subsequent two-layer frequency domain and depth feature fusion, depth features are extracted and downsampled using two different strides in the Bneck structure. The frequency-domain features are again processed with 2D DWT on the low-frequency information from the previous wavelet decomposition. These features are concatenated and fed into the next layer, where frequency and spatial domain features are continuously fused. The downsampling process in the network is replaced by 2D DWT along with dimensionality reduction operations to minimize the loss of valuable features. The network's tail retains the original MobileNetV3 structure, and Focal Loss [31] is used as the loss function to reduce the weight of easily classified samples, addressing issues related to sample imbalance. The formula for Focal Loss is as follows:

$$FL(p_t) = -\sum_{c=1}^{C} \alpha_c (1-p_c)^{\gamma} log(p_c)$$

(5)

Where *C* is the total number of categories, and $p_c$ is the model's predicted probability for category *c*.

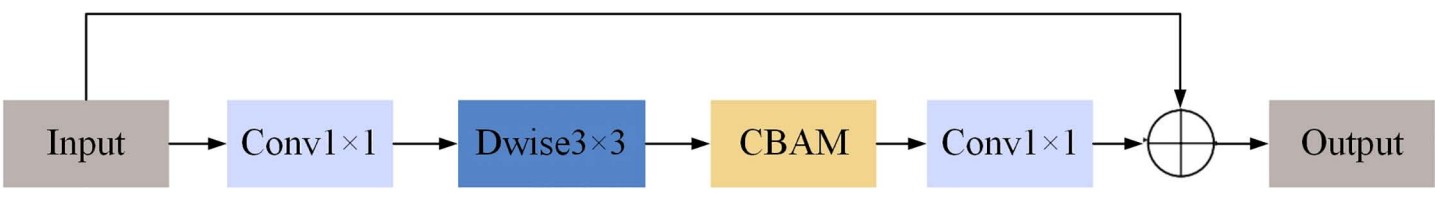

**Fig 6. Structure of Bneck.**

**Environment configuration and parameter setting.** The model construction and improvement are based on the TensorFlow 2.8.0 framework, with training and testing conducted on an Ubuntu 20.04 system. The server features an Intel Xeon E5-2690 V4 CPU and an NVIDIA GeForce RTX 3080Ti graphics card with 12 GB of memory for acceleration, utilizing CUDA version 11.4.0 and cuDNN 8.2.4.

The learning rate is a crucial factor influencing model performance; a learning rate that is too high can lead to better training set accuracy than test set accuracy, while a rate that is too low may prolong training time. The model was trained using learning rates of 0.005, 0.001, 0.0005, 0.0001, and 0.00005, with accuracy comparison curves illustrated in Fig 7. At a learning rate of 0.005, gradient explosion resulted in the highest training set accuracy of 24.25%. When the learning rate was set to 0.0005, the training set accuracy peaked at 98.70%. As the learning rate decreased, accuracy also declined. Considering convergence speed and accuracy, the optimal learning rate was 0.0005.

## Results and analysis

### Dataset quality assessment

The quality of the dataset significantly affects model performance. To examine the changes in dataset quality before and after data enhancement, the WaveLiteNet model was trained on both versions of the dataset, with the results illustrated in the curves shown in Fig 8.

Fig 8 shows that the original dataset's relatively small sample size leads to low model accuracy, increased overfitting, and poor recognition performance. In contrast, data enhancement significantly improves the accuracy of the trained model and accelerates convergence, effectively mitigating the overfitting issue. In summary, data enhancement yields high-quality datasets, enhancing the model's recognition accuracy while also helping to prevent overfitting and bolster its generalization capability.

### Experimental results of model structure optimization

The results of models with varying Bneck structures on the test set are shown in Table 2. The inverse residual structure in Bneck preserves the network's feature extraction capability. Models with Bneck structures (2, 2, 3, 2, 1) and (2, 2, 3, 4, 11–4) show the greatest parameter reduction compared to MobileNetV3. The (2, 3, 4, 11–4) model, while reducing parameters more than MobileNetV3, also maintains better accuracy than the (2, 2, 3, 2, 11–3) model. Therefore, the (2, 3, 4, 11–4) structure is chosen for fusion with the frequency domain features.

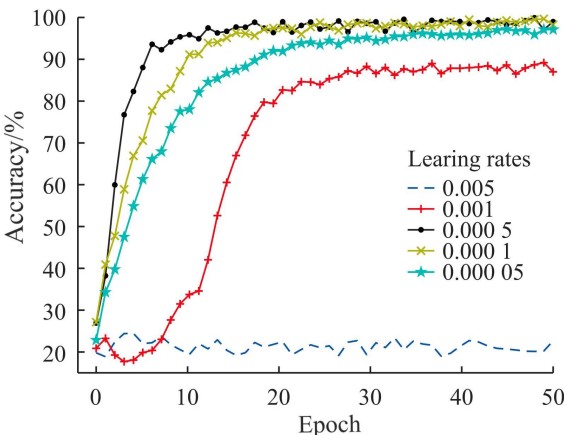

**Fig 7. Accuracy curves under different learning rates.**

The WaveLiteNet model was evaluated with varying numbers of deep feature fusion layers and frequency domains to determine the optimal structure for tea disease recognition. As shown in Table 2, adding more fusion layers slightly increased both parameters and FLOPs, significantly improving accuracy. Specifically, increasing the layers from 1 to 3 resulted in a 1.31% accuracy gain, a 0.93% increase in parameters, and 4.45% increase in FLOPs. While accuracy improved with more layers, further increases led to excessive parameters. Therefore, the optimal number of layers for the frequency domain and depth feature fusion model was determined to be 3.

Additionally, the recognition effectiveness varied with different channel ratios during feature fusion. Ratios of 0.5, 1.0, 1.5, and 2.0 were tested. Table 2 shows that a channel ratio of 1.0 achieved the highest accuracy, leading to double the number of channels in the fusion process.

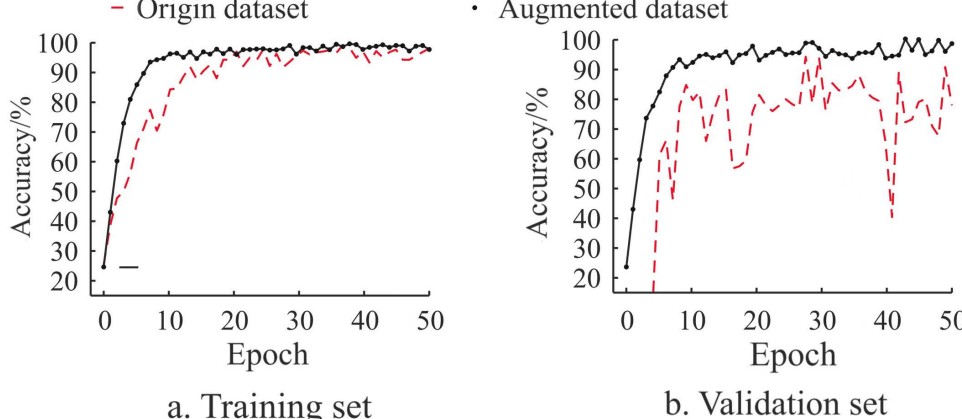

a. Training set    b. Validation set

**Fig 8. Comparison curves of accuracy.**

**Table 2. Results with model structure optimization.**

| Optimistic methods | Param settings | Accuracy/% | F1-Score/% | Parameters | FLOPs/10⁹ |
|---|---|---|---|---|---|
| Reduce Bneck structures | (2, 2, 3, 2, 11–3) | 96.12 | 96.00 | $2.91 \times 10^6$ | 0.31 |
| | (2, 2, 3, 2, 22,3) | 96.20 | 96.05 | $3.79 \times 10^6$ | 0.32 |
| | (2, 2, 3, 4, 11–4) | 96.42 | 96.12 | $3.51 \times 10^6$ | 0.35 |
| | (2, 2, 3, 4, 22–4) | 96.48 | 96.29 | $3.91 \times 10^6$ | 0.38 |
| | (2, 2, 3, 6, 22,3,6) | 96.59 | 96.58 | $4.28 \times 10^6$ | 0.44 |
| Adjust fusion layers | 1 | 97.41 | 97.82 | $3.13 \times 10^6$ | 0.43 |
| | 2 | 98.62 | 98.48 | $3.14 \times 10^6$ | 0.44 |
| | 3 | 98.72 | 98.69 | $3.18 \times 10^6$ | 0.46 |
| | 4 | 98.74 | 98.73 | $3.34 \times 10^6$ | 0.47 |
| Adjust fusion ratios | 0.5 | 97.90 | 97.86 | $3.15 \times 10^6$ | 0.44 |
| | 1.0 | 98.72 | 98.66 | $3.16 \times 10^6$ | 0.46 |
| | 1.5 | 98.69 | 98.48 | $3.18 \times 10^6$ | 0.48 |
| | 2.0 | 98.19 | 98.17 | $3.20 \times 10^6$ | 0.50 |

Note: FLOPs, floating-point operations.

## Model comparative experiment results

To evaluate the effectiveness of the frequency domain and deep feature fusion model, WaveLiteNet was compared with MobileNetV3, other lightweight recognition models, and existing tea disease recognition models using the tea disease dataset. Additionally, the impact of CBAM and Focal Loss was assessed by comparing WaveLiteNet with TealeafNet, which uses basic feature fusion. The comparison results across various evaluation metrics on the test set are shown in Table 3.

Overall, WaveLiteNet outperforms other models, achieving the highest average accuracy of 98.70% and excelling in F1 scores. While the model's FLOPs increased by 7.10% to $4.5 \times 10^8$ due to the fusion of frequency-domain and depth features, it achieved significant improvements in accuracy, precision, recall, and F1 score—rising by 2.15%, 2.24%, 2.15%, and 2.19%, respectively—while reducing parameters by 25.12% to $3.16 \times 10^6$.

Among other lightweight models, MCA-MobileNet, though having the lowest parameters, is effective in tea leaf disease recognition. However, MobileNet achieves accuracy below 95%, and LMA_CNNs exhibit slightly lower accuracy. The models in this study, particularly WaveLiteNet, demonstrate superior feature extraction, offering better accuracy and fewer parameters. WaveLiteNet improves accuracy by 0.62 percentage points over TealeafNet, showcasing the benefits of CBAM and Focal Loss.

WaveLiteNet also exhibits fewer misidentified samples compared to other models. Misidentifications in improved AlexNet and LeafNet often occurred due to background interference from other leaves and branches, while the SE-DenseNet-FL model struggled with noisy images. In contrast, WaveLiteNet effectively mitigates background and noise influences, proving its robustness.

A test set with small local occlusions showed WaveLiteNet's effectiveness, with an average accuracy of 98.30%. While Pseudocercospora disease images had the lowest recognition accuracy (above 96%), Cercospora Leaf Spot disease, with varying spot numbers, was correctly identified at an accuracy of 99.75%, demonstrating the model's resilience to small occlusions.

## Ablation experiment

To assess the impact of the Focal Loss function on model performance and explore the effect of various attention modules on recognition efficacy, we compared the TealeafNet model, which uses the cross-entropy loss function, with the same model using the Focal Loss function under different attention modules. The test results are presented in Table 4.

Table 3. Performance comparison with different models.

| Models | Accuracy/% | Precision/% | Recall/% | F1-Score/% | Parameters | FLOPs/10⁹ |
|---|---|---|---|---|---|---|
| MobileNetV3 | 96.55 | 96.45 | 96.52 | 96.48 | $4.22 \times 10^6$ | 0.42 |
| MCA-MobileNet [20] | 94.81 | 94.82 | 94.73 | 94.71 | $7.50 \times 10^5$ | 0.24 |
| LMA_CNNs [19] | 97.78 | 97.72 | 97.75 | 97.73 | $1.28 \times 10^7$ | 3.88 |
| VGG16 [4] | 97.34 | 97.40 | 97.32 | 97.32 | $1.38 \times 10^8$ | 15.48 |
| Improved AlexNet [10] | 92.13 | 92.18 | 91.94 | 91.98 | $1.70 \times 10^7$ | 0.18 |
| LeafNet [11] | 97.19 | 97.11 | 97.21 | 97.14 | $2.40 \times 10^7$ | 0.80 |
| MergeModel [13] | 97.56 | 97.61 | 97.64 | 97.60 | $3.74 \times 10^8$ | 16.27 |
| SE-DenseNet-FL [14] | 98.12 | 98.14 | 98.10 | 98.11 | $1.26 \times 10^7$ | 11.58 |
| TealeafNet | 98.06 | 98.02 | 98.05 | 98.03 | $3.05 \times 10^6$ | 0.41 |
| CBAM-TealeafNet | 98.70 | 98.69 | 98.6**7** | 98.67 | $3.16 \times 10^6$ | 0.45 |

The Focal Loss function offers significant advantages over cross-entropy for two main reasons:

1) Imbalanced dataset: The tea disease dataset has a sample imbalance, with diseases like Anthracnose and Blister Blight having fewer samples. This imbalance skews the model's optimization when using cross-entropy, as easily classified diseases dominate the overall loss.

2) Classification difficulty: Diseases like Pseudocercospora and Leaf Blight or Brown Blight have fewer samples, and the high similarity between these diseases, along with the small and dense distribution of Cercospora Leaf Spot, makes classification challenging. Focal Loss mitigates the impact of easily classified samples by adjusting hyperparameters, allowing the model to focus on more difficult cases during training, which enhances the model's optimization.

In the ablation study, we observed that CBAM outperformed both SENet and ECA in terms of accuracy. This improvement can be attributed to the distinctive way CBAM processes feature maps. SENet assigns weights to individual channels based on their importance, which is beneficial for general feature extraction. However, in the case of tea disease recognition, where features extracted through 2D DWT exhibit intricate interdependencies, SENet's simple channel-wise attention is insufficient to capture these complex relationships. ECA, on the other hand, focuses on enhancing channel attention without integrating spatial information, which limits its ability to process spatially varying disease features.

In contrast, CBAM integrates both spatial and channel attention mechanisms, enabling it to better capture the hierarchical and spatial relationships within disease features. By focusing on both the significant channels and relevant spatial regions, CBAM is more effective in filtering out irrelevant features and enhancing the representation of disease characteristics. This leads to improved performance in recognizing subtle disease patterns, which are critical in tea leaf disease classification.

Fig 9 shows the AUC-ROC curves for tea disease classification, further highlighting the robustness of the proposed model. The model achieved high performance in the Healthy category (AUC = 0.95) and in diseases like *Pseudocercospora* (AUC = 0.90) and *Anthracnose* (AUC = 0.89), confirming its ability to distinguish healthy from diseased tea. It also performed well in other categories, including *Cercospora Leaf Spot* (AUC = 0.85), *Blister Blight* (AUC = 0.83), and *Leaf Blight or Brown Blight of Tea* (AUC = 0.84), demonstrating strong generalization. Despite some lower AUC values, the model outperformed the random classifier (AUC = 0.5), suggesting its effective feature extraction.

This strong performance is attributed to the frequency domain and deep feature fusion strategy, which improves feature representation and minimizes background noise. The experimental results confirm that WaveLiteNet is efficient in tea disease classification and robust in complex scenarios, making it highly suitable for practical applications.

### Analysis of false identification reasons

To evaluate the performance improvement of WaveLiteNet over MobileNetV3, confusion matrices for both models were plotted based on the test set, as shown in Figs 10 and 11. The diagonal elements indicate correctly classified

**Table 4. Performance comparison under different loss functions and attention mechanisms.**

| Loss functions | Attention mechanisms | Accuracy/% | Precision/% | Recall/% | F1-Score/% |
|---|---|---|---|---|---|
| Cross Entropy | SENet | 98.06 | 98.02 | 98.05 | 98.03 |
| | ECA | 98.28 | 98.30 | 98.13 | 98.19 |
| | CBAM | 98.65 | 98.59 | 98.62 | 98.60 |
| Focal Loss | SENet | 98.24 | 98.21 | 98.15 | 98.18 |
| | ECA | 98.59 | 98.51 | 98.68 | 98.57 |
| | CBAM | 98.70 | 98.69 | 98.67 | 98.67 |

instances, with rows representing predicted labels and columns representing true labels.  shows that Mobile-NetV3 misclassifies many *Blister Blight* and *Anthracnose* cases as other diseases. In contrast, WaveLiteNet, by integrating frequency-domain and spatial-domain features, reduces misclassifications and improves robustness, particularly

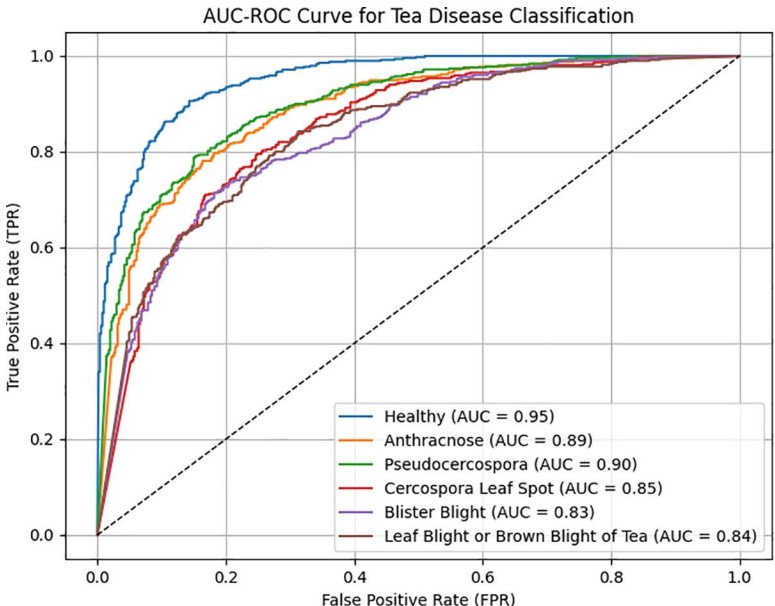

**Fig 9. AUC-ROC curve for tea disease classification.**

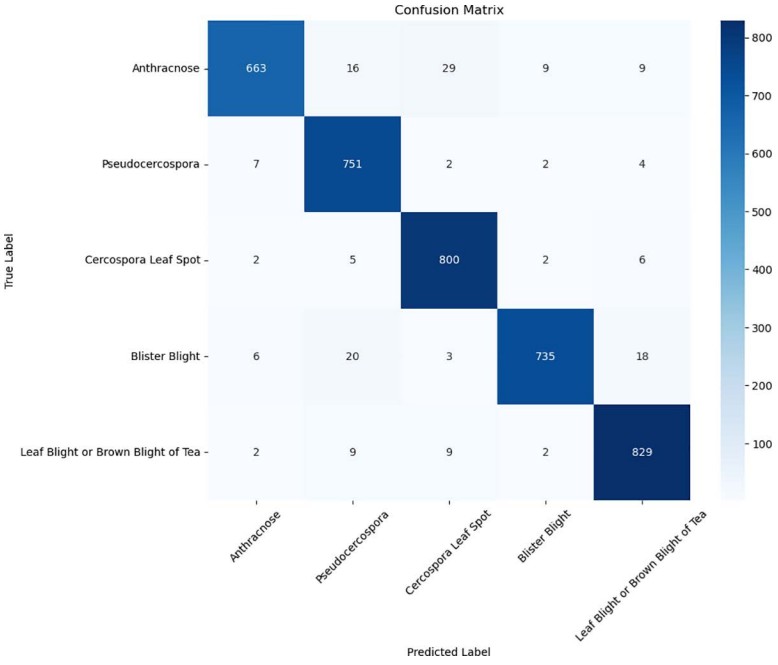

**Fig 10. Confusion matrix for MobileNetV3 on the test set.**

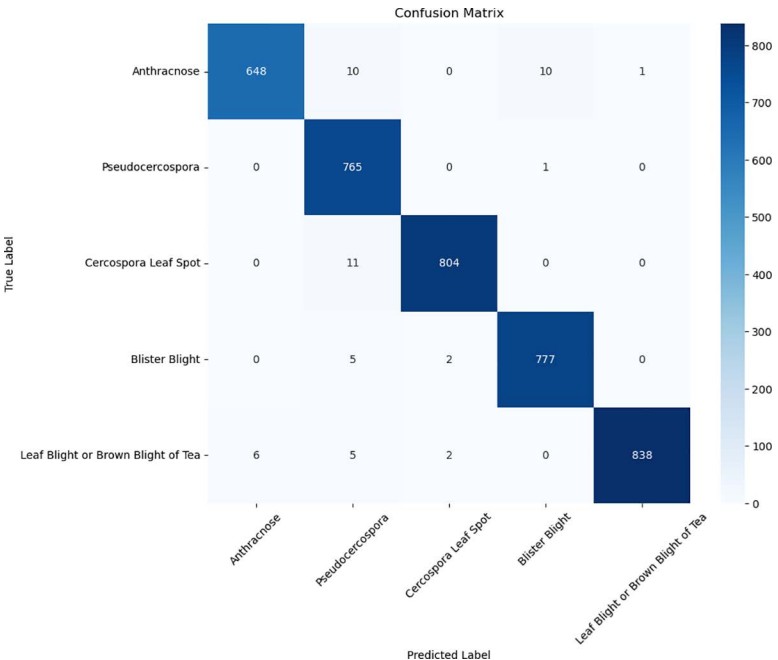

**Fig 11. Confusion matrix for WaveLiteNet on the test set.**

**Table 5. Comparison of WaveLiteNet and MobileNetV3 on low-resource devices.**

| Model | Inference time (ms) | Memory usage (MB) | Applicable devices |
|---|---|---|---|
| WaveLiteNet | 45 | 35 | ARM Architecture |
| MobileNetV3 | 85 | 80 | ARM Architecture |

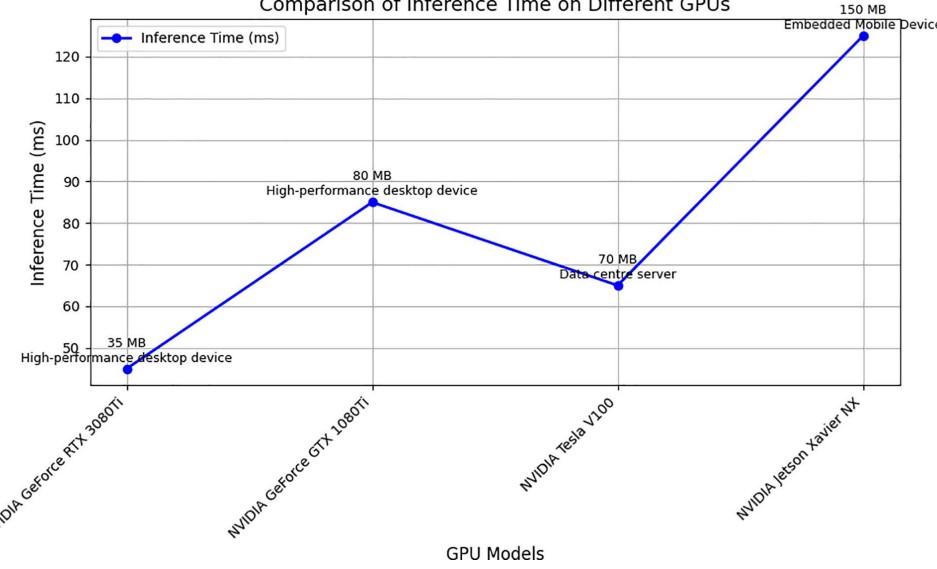

**Fig 12. Comparison of inference time on different GPUs.**

in distinguishing between foreground and background. However, it did misclassify some *Leaf Blight or Brown Blight* instances as *Cercospora Leaf Spot*, likely due to noise affecting texture extraction.

In Fig 11, ten *Anthracnose* images were misclassified as *Blister Blight*. These images appeared greyish-white due to brightness adjustments, making it challenging for the model to differentiate between the two diseases, especially when the leaf's edges mixed brown and greyish-white. Additionally, eleven images of *Leaf Blight or Brown Blight of Tea* were misclassified as *Cercospora Leaf Spot*, mostly due to noise and small spots resembling the characteristics of *Leaf Blight or Brown Blight*.

## Deployment comparison test results

This paper evaluates WaveLiteNet's processing speed, memory usage, and performance on low-resource devices to verify its performance in real deployment. The results are shown in Table 5.

Test results show that WaveLiteNet performs well on low-power devices and is capable of meeting real-time disease identification requirements.

## Appendix (Hardware Performance Comparison)

This paper compares inference times on different GPUs to further validate WaveLiteNet's performance on different hardware platforms. The test environment includes a variety of GPU configurations, and the results are shown in Fig 12.

This figure shows WaveLiteNet's inference time and memory footprint on different hardware configurations. Although the inference time is slightly longer on some low-power devices, the overall performance still meets the real-time requirements for tea disease identification, and the memory footprint is low, making it suitable for deployment on resource-constrained devices. Especially in mobile devices and embedded systems, WaveLiteNet still maintains high efficiency and provides a feasible solution for practical applications.

## Acknowledgments

This thesis is the culmination of a long and arduous journey that I have only been able to make because of the dedicated support I have received from many people.

The deepest and sincerest gratitude goes to my supervisor, associate Professor Gaojian Xu, for his continuous and invaluable guidance throughout my research. It is a great honor and privilege that I was allowed to work under his supervision. I would like to thank him for his patience, support, empathy, and great sense of humor. I am among the luckiest people who have benefited from Associate Professor Xu's knowledge and character. In addition, I am thankful to the academics and colleagues at the School of Information and Artificial Intelligence at Anhui Agricultural University. I would also like to thank my external and internal examiners for their valuable comments, which helped improve the thesis's quality. I would gratefully acknowledge the support of my dearest friends and research group mates for their encouragement, friendship, and support since the beginning of my study. I am extremely grateful to my parents for their motivation, patience, love, and care during my education and life journey. Without their help, this thesis would not have been accomplished. My final thanks go to all the people who have directly or indirectly supported me in completing my thesis. Sincere wishes to you. May everything be pleasant.

## Author contributions

**Conceptualization:** Jing Yang.

**Data curation:** MengDao Yang.

**Formal analysis:** ZhengPei Lin.

**Funding acquisition:** GaoJian Xu.

**Investigation:** MengDao Yang.

**Methodology:** Jing Yang.

**Project administration:** ZhengPei Lin.

**Resources:** GaoJian Xu.

**Software:** Jing Yang.

**Supervision:** GaoJian Xu.

**Validation:** Jing Yang.

**Visualization:** Jing Yang.

**Writing – original draft:** Jing Yang.

**Writing – review & editing:** Jing Yang, GaoJian Xu, MengDao Yang, ZhengPei Lin.

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
