## [Decision Letter · Decision Letter 0]

22 Dec 2024

PONE-D-24-43399Lightweight wavelet-CNN tea leaf disease detectionPLOS ONE

Dear Dr. Xu,

Thank you for submitting your manuscript to PLOS ONE. After careful consideration, we feel that it has merit but does not fully meet PLOS ONE’s publication criteria as it currently stands. Therefore, we invite you to submit a revised version of the manuscript that addresses the points raised during the review process.

We look forward to receiving your revised manuscript.

Kind regards,

Shahid Rahman, PhD

Academic Editor

PLOS ONE

2. Thank you for stating the following financial disclosure:  [Anhui Agricultural University Graduate Student Support Fund].  At this time, please address the following queries:

Additional Editor Comments (if provided):

Reviewers' comments:

Reviewer's Responses to Questions

**Comments to the Author**

1. Is the manuscript technically sound, and do the data support the conclusions?

Reviewer #1: Yes

2. Has the statistical analysis been performed appropriately and rigorously? 

Reviewer #1: Yes

3. Have the authors made all data underlying the findings in their manuscript fully available?

Reviewer #1: Yes

4. Is the manuscript presented in an intelligible fashion and written in standard English?

Reviewer #1: Yes

5. Review Comments to the Author

Reviewer #1: 1. The manuscript is technically sound and well-structured. The integration of 2D DWT, MobileNetV3, and CBAM is innovative and improves tea leaf disease detection. The methodology is rigorous, with appropriate dataset augmentation, architecture optimizations, and comparative studies. The results are robust, supported by strong statistical evidence.

2. Statistical analysis is appropriately performed using standard metrics (accuracy, precision, recall, F1-score). Ablation studies and comparative analyses validate improvements. The dataset splitting (6:2:2) and use of focal loss effectively address class imbalance.

3. Data is publicly accessible at https://github.com/jingyang-create/tea-sickness, meeting PLOS ONE’s data availability standards. The repository includes necessary materials for replication.

4. The manuscript is clear but requires minor proofreading for grammatical and typographical errors.

5. Recommendations for Improvement

1. Abstract Refinement:

• The content is detailed but could be improved by clearly separating key points to make it more accessible to a broader audience.

• Consider highlighting the practical applications at the beginning for better emphasis.

2. Introduction Depth:

• The literature review could be broadened to provide a clearer comparison of the advancements made by WaveLiteNet in relation to other lightweight models, such as SE-DenseNet-FL and TealeafNet.

3. Clarity in Methods:

• The section on the VisuShrink thresholding method is too technical and does not offer a sufficient explanation for readers who are not familiar with wavelet transformations.

• Include a simple diagram to illustrate the DWT process for better understanding.

4. Dataset Limitations:

• Figures 3, 4, and 5 are unclear and need better resolution for better understanding.

• It is not clear if data augmentation was only applied to the training set. If applied to the whole dataset, data leakage might occur. The authors should clarify this and explain how they addressed it.

5. Evaluation Metrics:

• Although comprehensive metrics are utilized, incorporating precision-recall curves or ROC curves could provide additional support for the claims related to the model's robustness.

6. Comparative Analysis:

• Expand the comparative analysis to cover real-world deployment scenarios or latency on mobile devices.

• While the authors propose that WaveLiteNet is suitable for deployment, they do not provide supporting evidence, such as information on processing speed, memory usage, or performance on low-resource devices.

7. Ablation Study Depth:

• The ablation study on CBAM could explore in greater detail why CBAM outperforms SENet or ECA in this particular use case.

8. Ethical Considerations:

• Although not a primary focus, mentioning ethical considerations regarding data usage, such as ensuring farmer consent for data collection, would improve transparency.

9. Writing Style:

• Certain sections, such as the Results and Analysis, are too verbose and could be condensed to improve readability without sacrificing important content.

10. Supplementary Data:

• Including a table or appendix that details the specific hardware performance, such as inference times on different GPUs, would be useful for readers interested in the feasibility of deployment.

6. PLOS authors have the option to publish the peer review history of their article (what does this mean? ). If published, this will include your full peer review and any attached files.

**Do you want your identity to be public for this peer review?** For information about this choice, including consent withdrawal, please see our Privacy Policy .

Reviewer #1: No

---

## [Author Response · Author response to Decision Letter 1]

3 Feb 2025

Dear Editor,

I would like to express my sincere gratitude for the reviewers' constructive feedback and the opportunity to revise my manuscript. I have carefully addressed all the comments and suggestions provided by the reviewers to improve the quality of the paper.

1)The ORCID iD in my personal information has been updated per your request.

2)I have provided a detailed explanation of the financial disclosure in the revised cover letter.

3)Regarding the issue of uploading figure files to the Preflight Analysis and Conversion Engine (PACE) digital diagnostic tool, I have followed the instructions and successfully uploaded my figures.

Thank you for your continued support and for considering my revised manuscript. I look forward to your feedback.

Dear Reviewer #1,

Thank you for your thoughtful and constructive feedback on our manuscript. We greatly appreciate your valuable suggestions, which have helped improve the quality of our paper. In response to your comments, we have made the following revisions:

1)Addressed methodological concerns – To provide greater clarity and reproducibility, we have refined the description of our research methodology, including the implementation details of the 2D DWT and Bneck structure.

2)Strengthened experimental validation—Additional explanations and statistical analyses have been incorporated to further support the effectiveness of our proposed approach.

3)Enhanced discussion and comparison—We have expanded the discussion section to provide a more comprehensive comparison with existing methods, emphasizing our model's advantages in terms of computational efficiency, robustness, and practical applicability.

4)Revised language and structure – The manuscript has been carefully edited to improve readability and ensure a more precise presentation of the research contributions.

These revisions have significantly enhanced the manuscript, and we sincerely hope the changes meet your expectations. Thank you again for your valuable insights and for helping us strengthen our work.

---

## [Editor Report · Decision Letter 1]

28 Feb 2025

PONE-D-24-43399R1Lightweight wavelet-CNN tea leaf disease detectionPLOS ONE

Dear Dr. Xu,

Thank you for submitting your manuscript to PLOS ONE. After careful consideration, we feel that it has merit but does not fully meet PLOS ONE’s publication criteria as it currently stands. Therefore, we invite you to submit a revised version of the manuscript that addresses the points raised during the review process.

We look forward to receiving your revised manuscript.

Kind regards,

Shahid Rahman, PhD

Academic Editor

PLOS ONE

**Journal Requirements:**

**Additional Editor Comments:**

1. Sections, Sub Sections are not in good format. Such Material and Method (1.,1.1, 1.1.1) etc.

2. Some Figures are not transparent such 3, 5 etc.

3. Follow the overall standard format of the Research article according to the PLOS ONE Criteria.

---

## [Author Response · Author response to Decision Letter 2]

11 Mar 2025

Dear Editor,

We sincerely appreciate the time and effort that you and the reviewers have devoted to evaluating our manuscript. We have carefully considered the comments and have made substantial revisions to address the concerns raised. Below is a summary of the modifications made in the revised manuscript:

Reference List Revision: We have thoroughly reviewed our reference list to ensure its completeness and accuracy. Based on the availability, relevance, and novelty of the cited works, we have replaced ten references with more appropriate and recent ones. Additionally, we have included DOI information for all references to facilitate accessibility for readers.

1、Reference list revision.

2、Manuscript formatting.

3、Figures improvement.

4、Overall formatting compliance.

Additionally, we confirm that no changes were required for the financial disclosure section.

We appreciate the constructive feedback from the reviewers and editors, which has helped us improve the quality of our manuscript. We hope the revised version meets the journal's publication criteria and look forward to your positive consideration.

Thank you for your time and consideration. We look forward to your further feedback. I appreciate your consideration.

Yours sincerely,

Jing Yang

---

## [Editor Report · Decision Letter 2]

6 Apr 2025

Lightweight wavelet-CNN tea leaf disease detection

PONE-D-24-43399R2

Dear Dr. Xu,

We’re pleased to inform you that your manuscript has been judged scientifically suitable for publication and will be formally accepted for publication once it meets all outstanding technical requirements.

Kind regards,

Shahid Rahman, PhD

Academic Editor

PLOS ONE
---

## [Editor Report · Acceptance letter]

PONE-D-24-43399R2

PLOS ONE

Dear Dr. Xu,

I'm pleased to inform you that your manuscript has been deemed suitable for publication in PLOS ONE. Congratulations! Your manuscript is now being handed over to our production team.

Kind regards,

on behalf of

Dr. Shahid Rahman

Academic Editor

PLOS ONE